# Resistance Training with Blood Flow Restriction Compared to Traditional Resistance Training on Strength and Muscle Mass in Non-Active Older Adults: A Systematic Review and Meta-Analysis

**DOI:** 10.3390/ijerph182111441

**Published:** 2021-10-30

**Authors:** Darío Rodrigo-Mallorca, Andrés Felipe Loaiza-Betancur, Pablo Monteagudo, Cristina Blasco-Lafarga, Iván Chulvi-Medrano

**Affiliations:** 1UIRFIDE (Sport Performance and Physical Fitness Research Group), Department of Physical and Sports Education, Faculty of Physical Activity and Sports Science, University of Valencia, 46010 Valencia, Spain; dariorodrigom@gmail.com (D.R.-M.); pmonteag@uji.es (P.M.); ivan.chulvi@uv.es (I.C.-M.); 2Department of Physical Education and Sports, Institute of Physical Education, University of Antioquia, Medellín 050010, Colombia; andres.loaiza@udea.edu.co; 3Department of Education and Specific Didactics, Jaume I University, 12071 Castellon, Spain

**Keywords:** hypertrophy, katsu, low-intensity training, occlusive exercise, sarcopenia

## Abstract

Low-intensity training with blood flow restriction (LI-BFR) has been suggested as an alternative to high-intensity resistance training for the improvement of strength and muscle mass, becoming advisable for individuals who cannot assume such a load. The systematic review aimed to determine the effectiveness of the LI-BFR compared to dynamic high-intensity resistance training on strength and muscle mass in non-active older adults. A systematic review was conducted according to the Cochrane Handbook and reportedly followed the PRISMA statement. MEDLINE, EMBASE, Web of Science Core Collection, and Scopus databases were searched between September and October 2020. Two reviewers independently selected the studies, extracted data, assessed the risk of bias and the quality of evidence using the GRADE approach. Twelve studies were included in the qualitative synthesis. Meta-analysis pointed out significant differences in maximal voluntary contraction (MVC): SMD 0.61, 95% CI [0.10, 1.11], *p* = 0.02, I^2^ 71% *p* < 0.0001; but not in the repetition maximum (RM): SMD 0.07, 95% CI [−0.25, 0.40], *p* = 0.66, I^2^ 0% *p* < 0.53; neither in the muscle mass: SMD 0.62, 95% CI [−0.09, 1.34], *p* = 0.09, I^2^ 59% *p* = 0.05. Despite important limitations such as scarce literature regarding LI-BFR in older adults, the small sample size in most studies, the still differences in methodology and poor quality in many of them, this systematic review and meta-analysis revealed a positive benefit in non-active older adults. LI- BFR may induce increased muscular strength and muscle mass, at least at a similar extent to that in the traditional high-intensity resistance training.

## 1. Introduction

The number of people over 60 years of age is increasing rapidly worldwide due to the increase in life expectancy and the decrease in the fertility rate. According to World Health Organization (WHO) data [1], the world population in this group of age is expected to reach 2 billion by 2050, reflecting an increase of 900 million from the 1.1 billion dated in 2015, up to the 22% of the total population compared to the current 12%. Maintenance of quality of life and prevention from disability (i.e., larger health span more than just life span), is of outermost importance and a current public health challenge [2].

In this scenery, physical activity (PA) has widely been confirmed to counteract the deterioration associated with aging and the sedentary behaviors intrinsic to these last stages of life [3,4]. PA reduces the risk of mortality and chronic pathologies [5,6]. It also helps to prevent dynapenia (decreased muscle strength) and sarcopenia (loss of strength, a decline in muscle mass, and final severe functional capacity impairment in the older adults, in this order) [7]. More specifically, physical exercise, especially strength training [8], emerges as a non-pharmacological tool in the management of this impairment of muscle function and structure which frequently leads older adults to the frailty syndrome [9]. Sarcopenia is indeed an emergency and expensive comprehensive health issue related to many other non-communicable diseases, such as larger number of falls and fractures [5,10,11], osteoporosis [12], diabetes [13], overall disability [12], but also cognitive impairment [14], reduced daily living autonomy, frequent hospitalization, and, finally, larger comorbidity and risk of death (See Cruz-Jentoft et al. [15], for Sarcopenia: revised European consensus on definition and diagnosis).

Resistance training is widely accepted as the most common strategy in this non-pharmacological approach to sarcopenia treatment [8]. Notwithstanding, in the last decade, research has revealed alternative training proposals to traditional high-intensity strength training (>70% RM), such as training with blood flow restriction (BFR), which consists of applying partial peripheral vascular occlusion during low-load strength training (20%–30% of 1RM), causing a local hypoxia effect in the muscle. Recent systematic reviews have analyzed responses on athletic population profiles [16,17] and active adults across the age spectrum (20–80 years) [18,19,20] indicating that BFR is a similarly effective intervention to high-intensity training in stimulating strength and muscle mass gains. Despite increasing research, the literature on BFR in older adults remains sparse on these issues and the subject’s functional status and moderating variables (pressure, cuff size, application volume), making further research necessary to strengthen the evidence on the efficacy of BFR in older adults.

Therefore, the present systematic review and meta-analysis aimed to determine the effectiveness of the low-intensity resistance training with blood flow restriction compared to dynamic high-intensity resistance training on strength and muscle mass in non-active older adults.

## 2. Materials and Methods

### 2.1. Protocol and Registration

This systematic review was conducted according to the Cochrane Handbook [21] and reported following the guidelines of the Preferred Reporting Items for Systematic Reviews and Meta-Analyses (PRISMA) declaration [22]. It was registered in the International Prospective Register of Systematic Reviews (PROSPERO registration number: CRD42020214901).

### 2.2. Information Sources and Search

We conducted a systematic search according to Chapter 4 of the Cochrane Handbook [21]. MEDLINE, Web of Science Core Collection, Scopus, and EMBASE databases were searched between September and October 2020. The search strategy applied was the following: old OR eld OR sarcopenic OR frail AND blood flow restriction OR occlusive training OR vascular occlusion OR kaatsu OR ischemic training.

### 2.3. Eligibility Criteria and Study Selection

Selection criteria were built based on the participants, intervention, comparators, outcomes, study design (PICOs) approach acronym [21] as follows.

Participants: Participants over 65 years, physically inactive, and characterized as healthy by the authors, defined as not achieving 150 min of moderate-to-vigorous-intensity physical activity per week or 75 min of vigorous-intensity physical activity per week or an equivalent combination of moderate and vigorous-intensity activity [23,24].

Intervention: Low-intensity blood flow restriction training (LI-BFR), based on the restriction of afferent and efferent blood flow during the performance of a low-intensity dynamic resistance exercise (20–50% of 1RM), causing a local hypoxia effect on the muscle using a pneumatic pressure cuff placed in the proximal region of the limb [25].

Comparators: Resistance training (RT) interventions were considered as any form of physical activity that is designed to improve muscular fitness by exercising a muscle or a muscle group against external resistance, performed systematically in terms of frequency, intensity, and duration, and is designed to maintain or enhance health-related outcomes. Resistance can come from fixed or free weights, elastic bands, body weight (against gravity), and water resistance. It may also involve static or isometric strength (holding a position or weight without moving against it). Often presented as a percentage of the participant’s one-repetition maximum (1-RM), the maximum weight they can lift/move if they only must do it once [26].

Outcomes: Muscular strength (Kg and Nm) and muscle mass (cm^2^).

Study design: Randomized controlled trial (RCT) where the intervention was RT with a follow-up period of at least 4 weeks. RCT is understood as a study in which many similar people are randomly assigned to 2 (or more) groups to test a specific drug, treatment, or other intervention. One group (the experimental group) has the intervention being tested, the other (the comparison or control group) has an alternative intervention, a sham dummy intervention (placebo), or no intervention at all. The groups are followed up to see how effective the experimental intervention was. Outcomes are measured at specific times and any difference in response between the groups is assessed statistically. This method is also used to reduce bias.

Eligibility criteria were applied independently by two blinded authors and disagreements were solved through consensus and active participation of a third author, likewise, the same authors inspected the reference lists from key journals and systematic reviews with a similar PICO to identified all promising or potential studies.

### 2.4. Data Collection Process

Two authors independently performed data extraction. Relevant data were extracted to a computer-based spreadsheet. The reviewers extracted the following information: authors’ information, publication year, functional status, BRFT characteristics (cuff size and pressure) resistance training protocols (frequency, intensity, length, duration, and volume), and effect estimates (mean, standard deviation, standard error) (Table 1).

### 2.5. Risk of Bias of Individual Studies

Two authors independently assessed the risk of bias. In the case of disagreement, the subject was discussed with another author. The risk of bias was assessed using the Cochrane risk-of-bias tool for randomized controlled trials (RoB 2.0) [39], which evaluates the risk of bias in five domains: the randomization process, deviations from intended interventions, missing outcome data, measurement of the outcome, and selection of the reported result. A study is considered to be at a “low risk of bias” if all five domains have been judged to be at low risk of bias. A study is considered to have “some concerns” if it has been judged to raise some concerns in at least one domain. A study is considered to be at a “high risk of bias” overall if it is judged to be at a high risk of bias in at least one domain. The tool was applied to each outcome of interest.

### 2.6. Summary Measures

For continuous outcomes, the group size, the mean values, and the standard deviations (SDs) were recorded for each group compared in the included studies. Pooled effects were calculated using an inverse of variance model, and the data were pooled to generate a standardized mean difference (SMDs) with a corresponding 95% confidence interval (CIs). Most studies for each outcome reported data in the same units, so it was possible to pool all studies regardless of whether they reported changes in-between data at baseline and final data. Significance was set at *p* < 0.05. A random-effects model was used. We used Cohen’s guidelines (no effect <0.2, small effect = 0.2 to 0.49, moderate effect = 0.5 to 0.79, large effect ≥ 0.80) [40] to report the magnitude of the effect and help with the interpretation of SMDs. All analyses were performed by a single reviewer using Review Manager (RevMan Version 5.4.1 The Cochrane Collaboration, 2020) and checked against the extracted data by the other author.

### 2.7. Additional Analysis

Subject to data availability, the subgroup analysis were performed considering the medium used to evaluate muscle strength and muscle mass on a specific muscle group or the evaluated kinetic chain.

### 2.8. Certainty of the Evidence: GRADE Approach

The reviewers decided *a posteriori* to evaluate the certainty of the evidence using the grading of recommendations, assessment, development, and evaluation (GRADE) approach to making the systematic review more usable for clinicians, trainers, decision-makers, and developers of clinical practice guidelines. We followed the GRADE approach to assess the certainty (or quality) of evidence in three major outcomes. The GRADE approach considers the risk of bias and the body of evidence to rate the certainty of the evidence into one of four levels:

High certainty: We are very confident that the true effect lies close to that of the estimate of the effect.

Moderate certainty: We are moderately confident in the effect estimate—the true effect is likely to be close to the estimate of the effect, but there is a possibility that it is substantially different.

Low certainty: Our confidence in the effect estimate is limited—the true effect may be substantially different from the estimate of the effect.

Very low certainty: We have very little confidence in the effect estimate—the true effect is likely to be substantially different from the estimate of effect.

## 3. Results

### 3.1. Literature Search and Article Selection

Initial database searches yielded a total of 1659 articles. After performing screening by title and abstract, and then removing duplicates, a total of 326 research papers were discarded, thus obtaining a total of 48 RCTs for full-text review. Subsequently, 36 RCTs were excluded for not assessing muscle mass and strength; apply BFR in aerobic exercise; results recorded on graphs only; apply BFR in pathological older adults. In total 12 studies were included in the Systematic Review [27,28,29,30,31,32,33,34,35,36,37,38] (Figure 1).

### 3.2. Risk of Bias Individual Studies

The twelve studies present some methodological problems.

#### 3.2.1. Muscular Strength Outcome (RM Test)

Four (57%) studies were judged of low risk of bias in at least one domain. One of them (14%) related to the random sequence generation and deviations from intended interventions [32]; three (43%) related to the missing data outcome [28,32,36]; and the remaining one (14%) for the measurement of the outcome domain [28]. For further information on the risk of bias, see Figure 2.

#### 3.2.2. Muscular Strength Outcome (MVC Test)

Four (57%) studies were judged of low risk of bias in at least one domain. Two of them (29%) were judged at low risk of bias in all domains [30,31]; four (57%) related to the missing data outcome [28,29,30,37], and the remaining three (43%) for the measurement of the outcome domain [28,29,30]. For further information on the risk of bias, see Figure 3.

#### 3.2.3. Muscle Mass Outcome (cm^2^)

Four (44%) studies were judged of low risk of bias in at least one domain. One of them (11%) related to the random sequence generation and deviations from intended interventions [31]; three (33%) related to the missing data outcome [29,36,37]; and the remaining one (11%) for the measurement of the outcome domain [28]. For further information on the risk of bias, see Figure 4.

### 3.3. Main Findings

#### 3.3.1. Narrative Synthesis

Twelve studies investigated the effect of the LI-BFR on strength and muscle mass compared to RT [27,28,29,30,31,32,33,34,35,36,37,38]. All studies that measured strength gains by direct RM test (kg) indicated significant improvements in weight lifted (*p* < 0.05) [28,31,32,33,34,36]. However, in the case of the studies that measured strength employing the MVC (Nm) [27,28,29,30,35,37,39], the evidence is a bit more uncertain, as two of the seven studies that performed this test did not find significant improvements in strength (*p* > 0.05) [27,35]. Table 2 describes the articles not included in the meta-analysis.

In the measurements concerning muscle mass of the included studies, those that measured changes in quadriceps thickness reported significant differences [27,28,31,33,34,36,38], however, for the lower limb, one study found no significant differences for adductors, hamstrings, and gluteus maximus [36]. In the case of upper extremities, one study reported significant differences in elbow flexor and extensor muscles [37].

#### 3.3.2. Quantitative Synthesis

The effects of BFR on muscular strength assessed in the RM-test, MVC-test, and muscle mass (cm^2^) are shown in Figure 5, Figure 6 and Figure 7, respectively.

##### LI-BFR vs. RT Alone on Muscular Strength via RM Test

As shown by Figure 5, when compared to resistance training alone, LI-BFR may have little to no effect in muscular strength measured by the RM test (SMD 0.07 (95% CI: −0.25 to 0.40) *p* = 0.66; I^2^ = 0%, *p* = 0.53). However, this evidence is very uncertain. Likewise, this evidence is very uncertain when analyzing this comparison separately in leg press, knee extension, and knee flexion (ES 0.01, ES 0.08, and 0.12, respectively; see Table 3).

##### LI-BFR vs. RT Alone on Muscular Strength via the MVC Testing

The LI-BFR effect on muscular strength measured using the MVC is larger than the one of RT alone (SMD 0.61, 95% CI [0.10 to 1.11], *p* = 0.02; I^2^ = 71%, *p* < 0.0001), but again, the evidence of this benefit is very uncertain.

Subgroup analysis by movement patterns reveals that this benefit is mainly due to the knee extension pattern, which is also significant (*p* = 0.05) and has a similar larger effect (SMD 0.65, 95% IC [0.00, 1.29]). Benefits in knee flexion are smaller and non-significant (SMD 0.53, 95% IC [−0.55, 1.61]; *p* = 0.33). Equally, there is also very uncertain evidence about this comparison on the MVC, both in the knee extension (ES 0.65), and in the knee flexion (ES 0.53). Table 4 shows this information.

##### LI-BFR vs. RT Alone on Muscle Mass (cm^2^)

Our data point out that LI-BFR trend to increase muscle mass over resistance training alone with a moderate effect size (SMD 0.62, 95% CI [−0.09 to 1.34], *p* = 0.09; I^2^ = 59%, *p* = 0.05), but the evidence is very uncertain (Figure 7). Likewise, the evidence is very uncertain about the effect of low-load BFR-RT when compared with RT alone on muscle mass in knee extensors (ES 0.26) and knee flexors (ES −0.20), and elbow flexors and extensors (ES 1.65), see Table 5.

## 4. Discussion

### 4.1. Summary of Main Results

Our review aimed to determine the effectiveness of the low-intensity resistance training with blood flow restriction compared to dynamic high-intensity resistance training on strength and muscle mass in non-active older adults. We included 6 randomized controlled trials in the meta-analysis, revealing that low-intensity blood flow restriction led to larger significant improvements in muscular strength (MVC test) compared to traditional resistance training. This larger benefit was reduced to a trend when considering the effect on the muscle mass (cross-sectional area, in cm) and even disappeared when comparing differences in muscular strength improvements assessed utilizing the RM test. Particularly, all these outcomes shared a very low level of certainty due to poor quality study designs and disparities in the methodological approach.

Notably, subgroup analysis by movement patterns revealed that the above-mentioned benefit on muscular strength assessed utilizing the MVC was mainly due to the knee extension pattern.

### 4.2. Certain of Evidence

The included studies evaluated different resistance training programs with or without BFR. The protocols in these studies differed in terms of the number of sets and repetitions, exercises, and muscle groups involved, as well as in the level of occlusion cuff pressure. Their positions regarding the characteristics of the participants, more specifically on the functional status, were neither entirely clear, as they previously justify the use of BFR in older adults with sarcopenia, yet no information on specific diagnostic tests for sarcopenia was found [15]. Moreover, the functional status of the subjects was determined as inactive (more than 6 months without physical activity), but older adults are a highly heterogeneous population [41], and their exercise-response is also heterogeneous [42], which needs further details. Therefore, the articles included in this review lack clear and unified criteria in the process of sample selection.

Very low-quality evidence formed all the comparisons in this systematic review. Our certainty in the evidence was downgraded due to limitations in the risk of bias assessment, including lack of both randomization process, measurement of the outcome, and selection of the reported result. The absence of blinding of both participants and investigators can lead to an overestimation of the effect estimate, although in exercise interventions it is not easy to blind participants. Of outermost importance, this blinding process is even more difficult in protocols with blood flow restriction, since if familiarization with the device and prior measurement of arterial occlusion pressure with Doppler ultrasound (which all the included studies affirm) have been properly conducted, it is easy to know whether the cuff is exerting pressure on the involved limb. Blinding the intensity is a challenge. Furthermore, most of the studies had low numbers of participants, wide confidence intervals, and high heterogeneity in the effects across them. Importantly, undertaking a sensitivity analysis to explore these limitations was not appropriate due to the low number of studies, which could bias any effect estimate.

### 4.3. Potential Biases in the Review Process

The strength of this systematic review was the use of systematic methods to assess the certainty of the evidence. An important limitation in the review process has been, as mentioned above, the heterogeneity of the training and BFR protocols.

Regarding strength training protocols, the number of repetitions was very disparate among the included studies with a range between 6 and 30 repetitions, including one study on muscle failure [32]. This high heterogeneity makes a comparison between studies difficult because the influence of the exercise program on the BFR effect cannot be completely isolated.

Another example of the heterogeneity of the protocols is the occlusion pressure. The included studies used different pressure percentages within the range established by the current positionings [43], and the pressure was calculated in two different ways. Some studies used Doppler ultrasound to determine the maximum arterial occlusion pressure while others applied a pressure value 1.5 times the brachial systolic pressure. It also happens that some studies used variable occlusion protocols (no pressure exerted in the recovery periods between series) while the rest were based on a constant pressure during the entire intervention, making direct comparisons between the results of the studies difficult.

### 4.4. Agreements and Disagreements with Other Studies or Reviews

Our findings of low-quality evidence on the effects of BFR on strength and muscle mass align with those reported by two recent systematic reviews [20,44]. For instance, the increase in muscle strength was revealed with effect sizes ranging from 0.55 to 4.34 [44]. Aligned with it, Centner et al. [20] found a greater improvement in muscle strength with pooled effect sizes (ES) of 2.16 (95% CI 1.61 to 2.70). These authors also highlighted a very low level of evidence for the included studies due to the methodological diversity and the very small sample of participants. They included profiles of unhealthy subjects, and they also reported the variability of the profiles due to the high heterogeneity of the elderly. Similarly to us, these reviews revealed a favorable trend for LI-BFR in muscle mass gain, however, this effect did not reach statistical significance. Since the methodological diversity of the above-cited primary reviews [20,44] is similar to ours, we may conclude that the profile and heterogeneity of the physical condition of the participants, being in this systematic review and meta-analysis of older adults, may influence the results regarding LI-BFR resistance training. The agreement between their findings and ours could be also explained by several factors like the control of other important variables, such as the nutritional status [45].

### 4.5. Implications for Practice and Further Research

The findings of this systematic review highlight the need for more RCTs, but mostly with a more defined methodological approach in their interventions, since the disparity of the protocols is detrimental to the quality of the evidence, as determined by the grading of recommendations, assessment, development, and evaluation (GRADE). In addition, all the primary studies included, together with those found in other systematic reviews, analyze muscular strength gains through specific strength tests, but there is a lack of knowledge about the effect of LI-BFR on the functional status of the elderly. Of course, it has been previously proven that increasing strength and muscle mass benefits physical capacity in older adults [4,8], but future lines of research might include together strength testing some functional assessments, or even some specific motor tasks and challenges of daily living activities, to determine the impact of BFR on functional capacity and older individuals’ autonomy.

## 5. Conclusions

The findings of this systematic review point out that strength training with blood flow restriction may induce increased muscular strength and muscle mass in non-active older adults, at least at a similar extent to that in the traditional high-intensity resistance training. However, caution should be when considering these findings, since the evidence is very uncertain about the effect of low-load BFR-RT when compared with RT alone on our outcomes. Further randomized controlled trials with a more defined and standardized methodological protocol are still required and more research is needed to reach a more certain conclusion.

## Figures and Tables

**Figure 1 ijerph-18-11441-f001:**
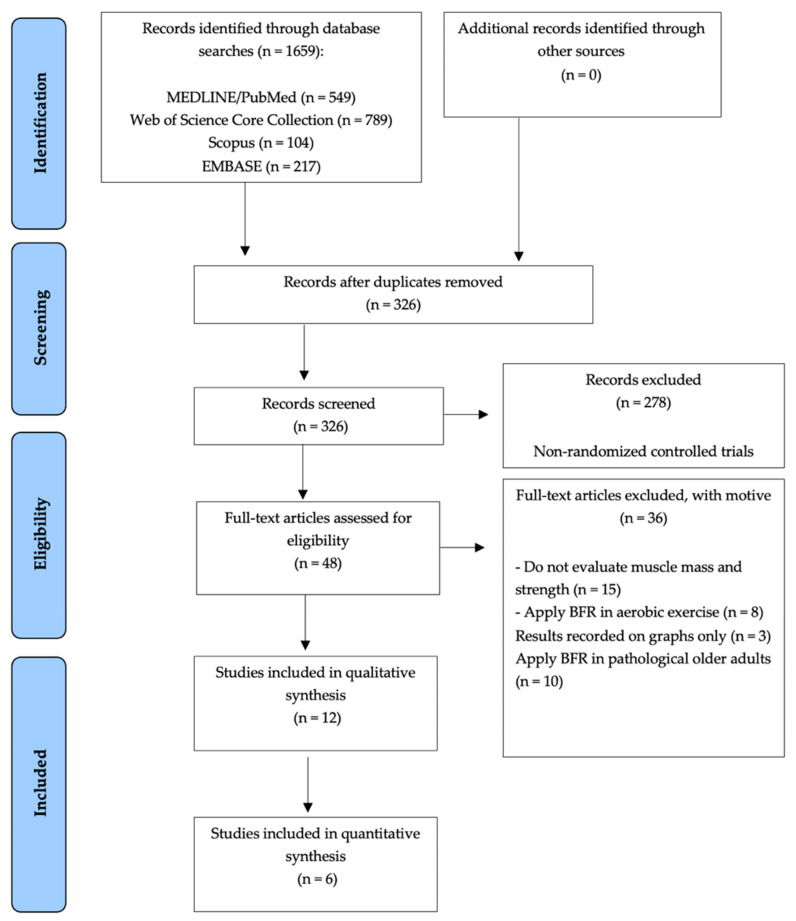
Preferred reporting items for systematic reviews and meta-analyses (PRISMA) flow-chart of the study selection.

**Figure 2 ijerph-18-11441-f002:**
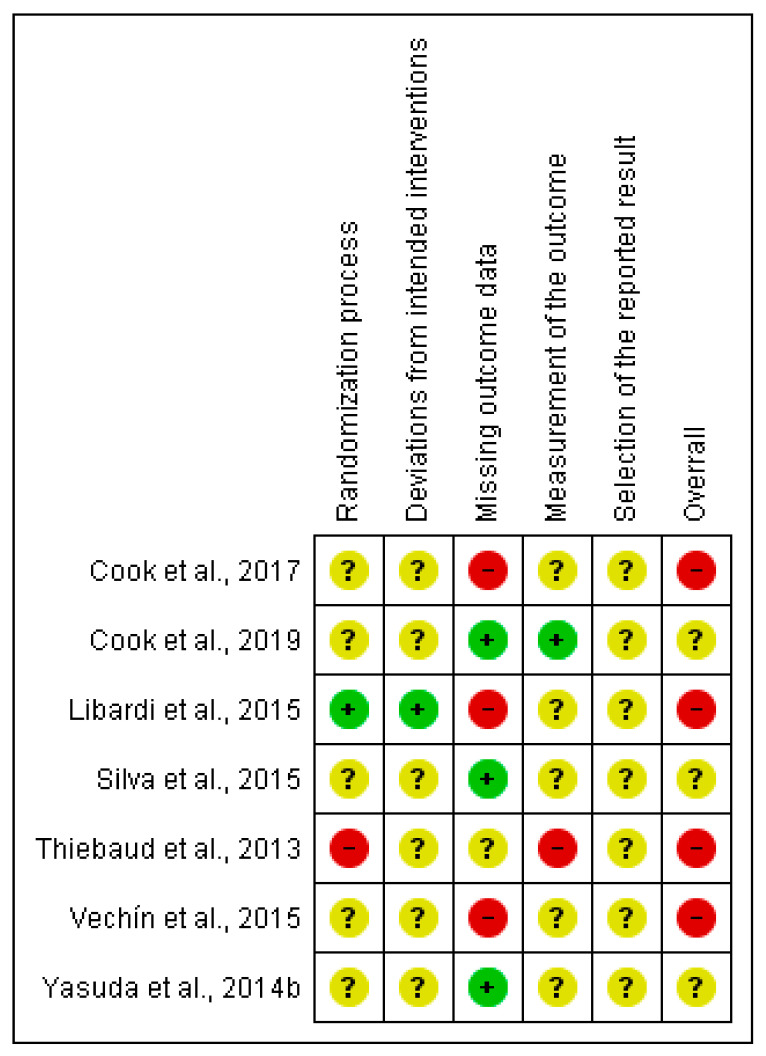
Risk of bias summary: review authors’ judgments about each risk of bias item for muscular strength outcome.

**Figure 3 ijerph-18-11441-f003:**
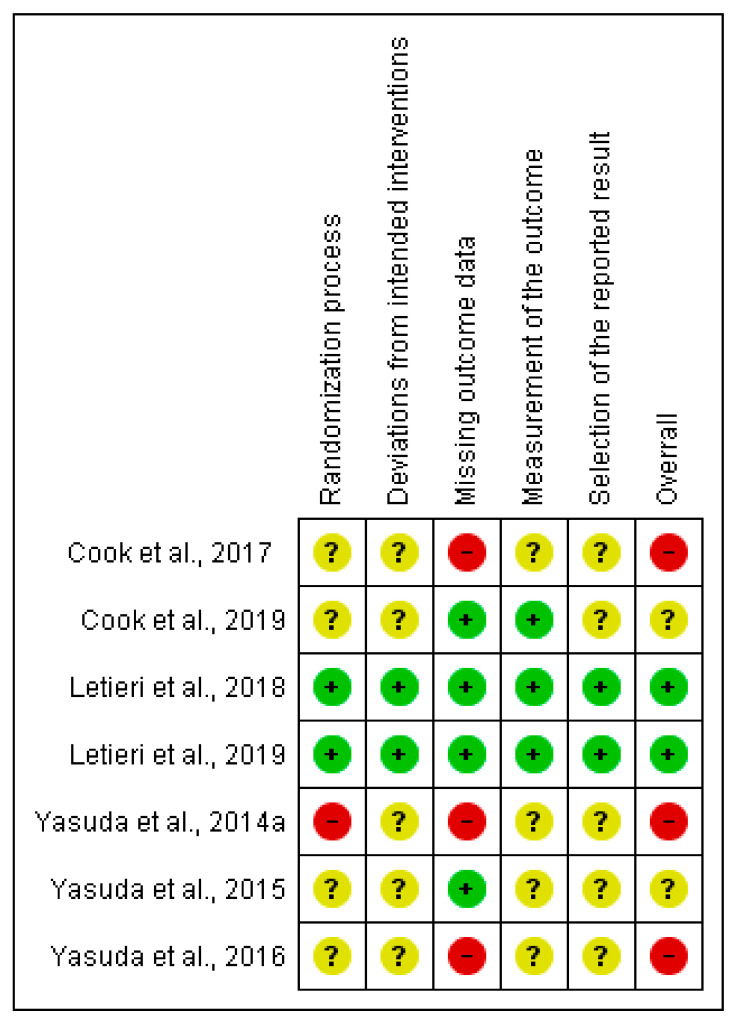
Risk of bias summary: review authors’ judgments about each risk of bias item for muscular strength outcome.

**Figure 4 ijerph-18-11441-f004:**
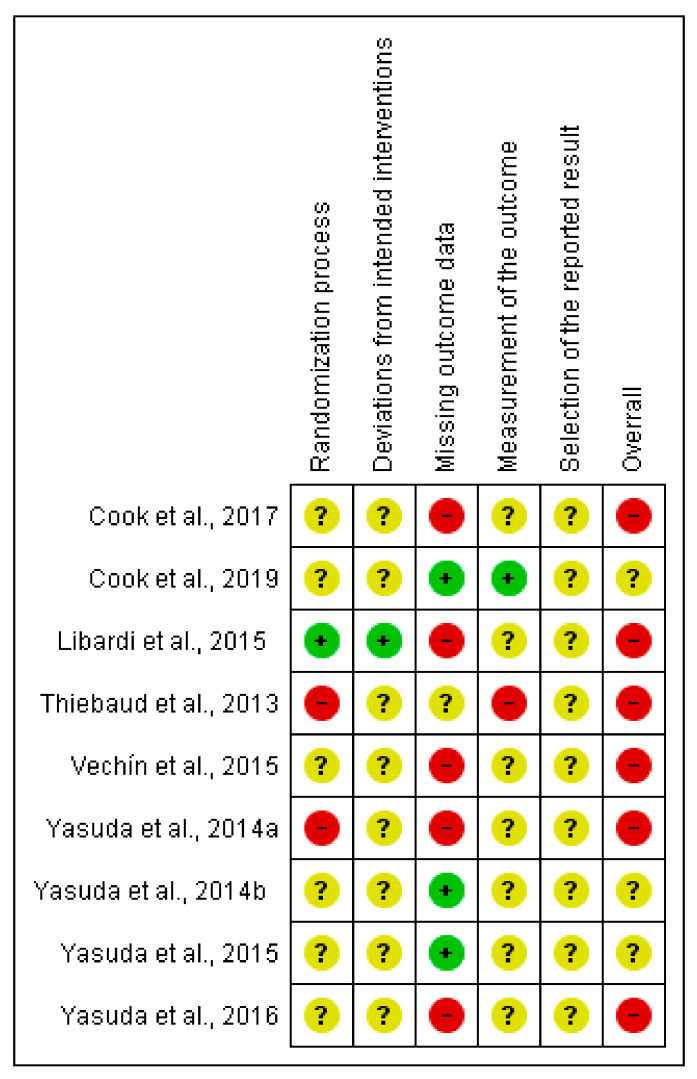
Risk of bias summary: review authors’ judgments about each risk of bias item for muscle mass outcome.

**Figure 5 ijerph-18-11441-f005:**
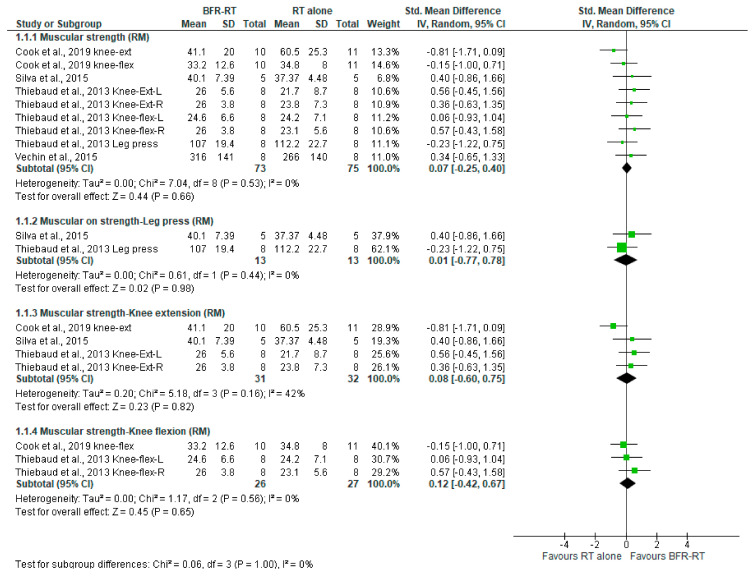
LI-BFR versus RT on muscular strength (RM test), standard means difference (SMD).

**Figure 6 ijerph-18-11441-f006:**
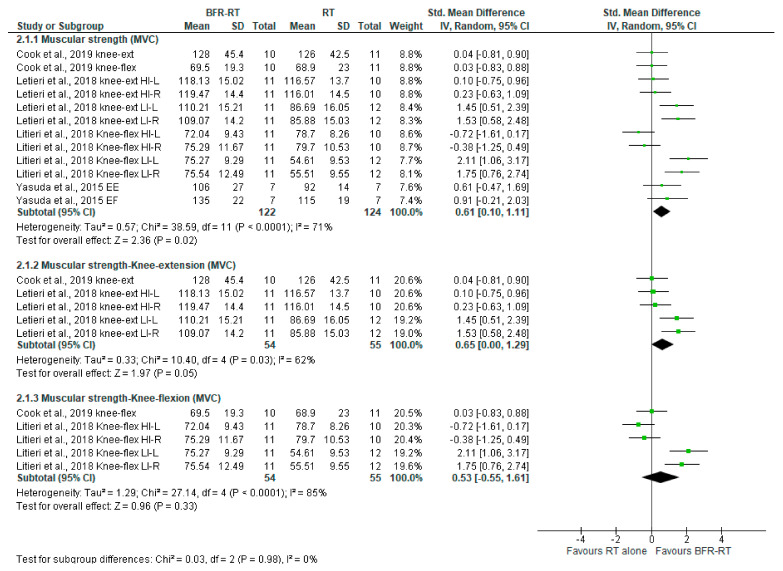
LI-BFR versus RT on muscular strength (MVC test), standard means difference (SMD).

**Figure 7 ijerph-18-11441-f007:**
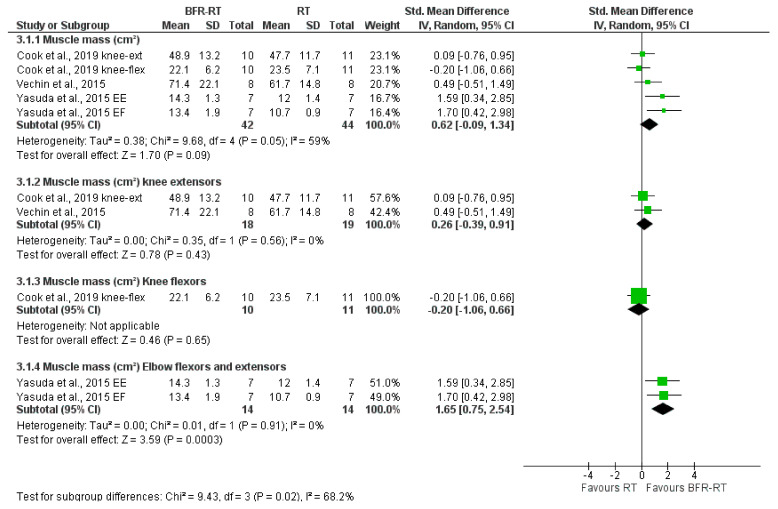
LI-BFR versus RT on muscle mass, standard means difference (SMD).

**Table 1 ijerph-18-11441-t001:** Characteristics of included studies.

Study	N	Age (yrs)	Functional Status	% 1RM	Cuff (cm)	Pressure (mmHg)	Frequency (d/wk)	Duration (wk)	Protocol (st × rp)	Measurements
Cook et al., 2017 [27]	36	69–82	Non active and risk of functional limitation	RT: 70% LI-BFR: 30% (LE and LC) and 50% (LP)	6	184 ± 25	2	12	RT: 3 × 10 LI-BFR: 3 × 10	CSA-MRI; MVC
Cook et al., 2019 [28]	21	67–85	Non active and risk of functional limitation	RT: 70% LI-BFR: 30% (LE and LC) and 50% (LP)	6	184 ± 25	2	12	RT: 3 × 10 LI-BFR: 3 × 10	CSA-MRI; MVC; 10RM Test
Letieri et al., 2018 [29]	56	68.8 ± 5.09	Non active	RT: 80% BFR: 30%	Not stated	BFRH: 185 ± 5 BFRL: 105 ± 6	3	16	RT: 3 × 6–8 LI- BFR: 1 × 30 + 3 × 15	MVC
Letieri et al., 2019 [30]	23	69.4 ± 5.73	Non active	RT: 80% BFR: 30%	13	80%	3	16	RT: 3 × 6–8 LI-BFR: 1 × 30 + 3 × 16	AMM; MVC
Libardi et al., 2015 [31]	25	64.7 ± 4.1	Non active	RT: 80% BFR: 20%	18	50%	3	12	RT: 4 × 10 LI-BFR: 1 × 30 + 3 × 15	CSA-MRI; RM Test
Silva et al., 2015 [32]	15	61.8 ± 6.01	Non active	RT: 80% BFR: 30%	18	80%	2	12	RT: 4 × Fail LI-BFR: 4 × Fail	RM Test
Thiebaud et al., 2013 [33]	14	60.5 ± 3.5	Non active	RT: 80% BFR: 30%	18	80-120	3	8	RT: 3 × 10 LI-BFR: 1 × 30 + 2 × 15	CSA-MRI; RM Test
Vechin et al., 2015 [34]	23	59–71	Non active	RT: 80% BFR: 30%	18	50%	2	12	RT: 4 × 10 LI-BFR: 1 × 30 + 3 × 15	CSA-MRI; RM Test
Yasuda et al., 2014 (a) [35]	17	61–85	Non active	Not stated	3	196 ± 18	2	12	RT: 4 × 10 LI-BFR: 1 × 30 + 3 × 16	CSA-MRI; MVC
Yasuda et al., 2014 (b) [36]	16	61–78	Non active	Not stated	Not stated	120–270	2	12	RT: 4 × 10 LI-BFR: 1 × 30 + 3 × 17	CSA-MRI; 10RM Test
Yasuda et al., 2015 [37]	14	61–85	Non active	Not stated	Not stated	202 ± 8	2	12	RT: 4 × 10 LI-BFR: 1 × 30 + 3 × 18	CSA-MRI; MVC
Yasuda et al., 2016 [38]	30	61–86	Non active	Not stated	5	160–200	2	12	RT: 3 × 12 LI-BFR: 1 × 30 + 3 × 15	CSA-MRI; MVC

Abbreviations: yrs, years; RT, resistance training exercise group; LI-BFR, low-intensity blood flow restriction exercise group; d, days; wk, week; st, sets; RP, repetitions; LC, leg curl; LE, leg extension; LP, leg press; CSA, cross-sectional area; MRI, magnetic resonance imaging; MVC, maximal voluntary contraction.

**Table 2 ijerph-18-11441-t002:** Data from studies not included in the meta-analysis.

Study ID	Population	Intervention	Comparison	Outcome		
				LI-BFR (Mean/PI ± SD)	RT (Mean/PI ± SD)	CON (Mean/PI ± SD)
Cook et al. (2017) - United States	36 elderly males and females non-active and risk of functional limitation with ages between 69 and 82 years	LI-BFR (n = 12)	RT and stretching (CON): RT (n = 12) CON (n = 12)	LE (RM-kg): 9.1, 95% CI [5, 13.2] *p* < 0.01 LC (RM-kg): 5.4, 95% CI [0.5, 10.2] *p* < 0.01 LP (RM-kg): 18.7, 95% CI [9.0, 28.4] *p* < 0.01 MVC (Nm): 11.2, 95% CI [-2.7, 25] *p* = 0.14 CSA (cm^2^): 3.23, 95% CI [1.29, 5.16] *p* < 0.01	LE (RM-kg): 21.2, 95% CI [13, 29.5] *p* < 0.01 LC (RM-kg): 8.2, 95% CI [5.4, 11.1] *p* < 0.01 LP (RM-kg): 31.7, 95% CI [13.6, 50] *p* < 0.01 MVC (Nm): 19.3, 95% CI [8.3, 30.3] *p* = 0.14 CSA (cm^2^): 2.86, 95% CI [1.87, 3.86] *p* < 0.01	LE (RM-kg): 0.6, 95% CI [−4.2, 5.3] *p* < 0.01 LC (RM-kg): 0.4, 95% CI [−1, 1.8] *p* < 0.01 LP (RM-kg): −0.2, 95% CI [−10.4, 10.1] *p* < 0.01 MVC (Nm): 3.5, 95% CI [−7.3, 14.3] *p* = 0.14 CSA (cm^2^): 0.07, 95% CI [−0.67, 0.82] *p* < 0.01
Letieri et al. (2019) - Brazil	56 elderly females non-active with ages between 63 and 74 years	LI-BFR (n = 11)	RT (n = 12)	HG (kg): 23.02 ± 3.2, *p* = 0.432	HG (kg): 23.04 ± 5.97, *p* = 0.432	No control group
Libardi et al. (2015) - Brazil	25 elderly males and females non-active with ages between 60 and 69 years	LI-BFR (n = 10)	RT and other unspecified (CON): RT (n = 8) CON (n = 7)	Percent increase (PI) Strength (RM-kg): 35.4%, *p* = 0.001 CSA (cm^2^): 7.6%, *p* < 0.0001	Percent increase (PI) Strength (RM-kg): 38.1%, *p* < 0.001 CSA (cm^2^): 7.3%, *p* < 0.0001	Percent increase (PI) Strength (RM-kg): −4.3%, *p* > 0.05 CSA (cm^2^): −2.2%, *p* > 0.05
Yasuda et al. (2014) (a) - Japan	17 elderly males and females non-active with ages between 61 and 85 years	LI-BFR (n = 9)	RT (n = 8)	Percent increase (PI) EF (MVC-Nm): 7.8%, *p* = 0.0082 EE (MVC-Nm): 16.1%, *p* = 0.0131 EF (CSA-cm^2^): 17.6%, *p* < 0.0001 EE (CSA-cm^2^): 17.4%, *p* = 0.0131	Percent increase (PI) EF (MVC-Nm): No changes EE (MVC-Nm): No changes EF (CSA-cm^2^): No changes EE (CSA-cm^2^): No changes	No control group
Yasuda et al. (2014) (b) - Japan	16 elderly males and females non- active with ages between 61 and 78 years	LI-BFR (n = 8)	RT (n = 8)	LE (RM-kg): 66 ± 27, *p* < 0.01 LP (RM-kg): 191 ± 60, *p* < 0.01 QD (CSA-cm^2^): 49.1 ± 9.6, *p* < 0.01 AD (CSA-cm^2^): 24.2 ± 8.4, *p* > 0.05 HM (CSA-cm^2^): 22.1 ± 4.8, *p* > 0.05 GM (CSA-cm^2^): 40.8 ± 7, *p* = 0.07	LE (RM-kg): 63 ± 24, *p* > 0.05 LP (RM-kg): 158 ± 44, *p* > 0.05 QD (CSA-cm^2^): 44.7 ± 8.9, *p* > 0.05 AD (CSA-cm^2^): 20.8 ± 3.6, *p* > 0.05 HM (CSA-cm^2^): 20.8 ± 3.6, *p* > 0.05 GM (CSA-cm^2^): 36.5 ± 7.7, *p* > 0.05	No control group
Yasuda et al. (2016) - Japan	30 elderly females non- active with ages between 61 and 86 years	LI-BFR (n = 10)	RT and other unspecified (CON): RT (n = 10) CON (n = 10)	Percent increase (PI) Strength (RM-kg): 16.4%, *p* < 0.001 Strength (MVC-Nm): 13.7%, *p* = 0.028 CSA (cm^2^): 6.9%, *p* < 0.001	Percent increase (PI) Strength (RM-kg): 17.6%, *p* < 0.001 Strength (MVC-Nm): No changes, *p* = 0.196 CSA (cm^2^): 1.5%, *p* = 0.871	Percent increase (PI) Strength (RM-kg): No changes *p* = 0.912 Strength (MVC-Nm): No changes, *p* = 0.810 CSA (cm^2^): −2.2%, *p* = 0.395

Abbreviations: LI-BFR, Low-intensity blood flow restriction exercise group; RT, resistance training exercise group; CON, control group; LE, leg extension; LC, leg curl; LP, leg press; HG, handgrip; EF, elbow flexion; EE, elbow extension; QD, quadriceps; AD, adductors; HM, hamstrings; GM, gluteus maximus, CI, confidence interval; PI, percent increase; SD, standard deviation.

**Table 3 ijerph-18-11441-t003:** Summary of findings for the comparison: LI-BFR versus RT alone on muscular strength (RM test).

Resistance Training with Blood Blow Restriction Versus Resistance Training Alone					
**Population:** Non-Active older adults **Intervention:** resistance training with blood flow restriction **Comparison:** resistance training **Setting:** laboratory					
**Outcomes**	**Relative Effect (95% CI)**	**Anticipated Absolute Effect *** **(95% CI)**		**N° of** **Participants** **(Studies)**	**Certainty** **of the Evidence** **(Grade)**
		**Assumed Risk with Control**	**Assumed Risk with Intervention**		
**Muscular strength (RM Test)** Up to 12 weeks	SMD 0.07 * (−0.25 to 0.40)	21.7 to 266	Mean strength in intervention was 0.07 higher (0.25 lower to 0.40 higher)	148 (4 RCTs)	⨁◯◯◯ VERY LOW ^1,2^
**Muscular strength-Leg press (RM Test)** Up to 12 weeks	SMD 0.01 * (−0.77 to 0.78)	37.37 to 112.2	Mean strength in intervention was 0.01 higher (0.77 lower to 0.78 higher)	26 (2 RCTs)	⨁◯◯◯ VERY LOW ^1,2^
**Muscular strength-Knee extension (RM Test)** Up to 12 weeks	SMD 0.08 * (−0.60 to 0.75)	21.7 to 60.5	Mean strength in intervention was 0.08 higher (0.60 lower to 0.75 higher)	63 (3 RCT)	⨁◯◯◯ VERY LOW ^1,2^
**Muscular strength-Knee flexion (RM Test)** Up to 12 weeks	SMD 0.12 * (−0.42 to 0.67)	23.1 to 34.8	Mean strength in intervention was 0.12 higher (0.42 lower to 0.67 higher)	53 (2 RCTs)	⨁◯◯◯ VERY LOW ^2,3^

The risk in the intervention group (and its 95% confidence interval) is based on the assumed risk in the comparison group and the relative effect of the intervention (and its 95% CI). CI: Confidence interval; RM: maximum repetitions; SMD: Standard mean difference. ***** Effects size: 0.2 represents a small effect, 0.5 a moderate effect, and 0.8 a large effect [29]. ^1^ Downgraded by two levels due to no randomization process, selection of the reported result, and measurement of the outcome. ^2^ Downgraded by two-level due to small sample size and wide confidence intervals (imprecision); ^3^ Downgraded by one level due to no randomization process, and selection of the reported result.

**Table 4 ijerph-18-11441-t004:** Summary of findings for the comparison: LI-BFR versus RT alone on muscular strength (MVC test).

Resistance Training with Blood Blow Restriction Versus Resistance Training					
**Population:** Non-active older adults **Intervention:** resistance training with blood flow restriction **Comparison:** resistance training **Setting:** laboratory					
**Outcomes**	**Relative Effect (95% CI)**	**Anticipated Absolute Effect *** **(95% CI)**		**N° of** **Participants** **(Studies)**	**Certainty** **of the Evidence** **(Grade)**
		**Assumed Risk with Control**	**Assumed Risk with Intervention**		
**Muscular strength (MVC test)** Up to 16 weeks	SMD 0.61 * (0.10 to 1.11)	54.61 to 126	Mean strength in intervention was 0.61 higher (0.10 lower to 1.11 higher)	246 (3 RCTs)	⨁◯◯◯ VERY LOW ^1,2^
**Muscular strength-Knee extension (MVC test)** Up to 16 weeks	SMD 0.65 * (0.00 to 1.29)	85.88 to 126	Mean strength in intervention was 0.65 higher (0.00 lower to 1.29 higher)	109 (2 RCTs)	⨁◯◯◯ VERY LOW ^1,2^
**Muscular strength-Knee flexion (MVC test)** Up to 16 weeks	SMD 0.53 * (−0.55 to 1.61)	85.88 to 126	Mean strength in intervention was 0.53 higher (0.55 lower to 1.61 higher)	109 (2 RCTs)	⨁◯◯◯ VERY LOW ^1,2^

The risk in the intervention group (and its 95% confidence interval) is based on the assumed risk in the comparison group and the relative effect of the intervention (and its 95% CI). CI: Confidence interval; MVC: Maximum voluntary contraction; SMD: Standard mean difference. ***** Effects size: 0.2 represents a small effect, 0.5 a moderate effect, and 0.8 a large effect [29]. ^1^ Downgraded by one level due to inconsistency; ^2^ Downgraded by two-level due to small sample size and wide confidence intervals (imprecision).

**Table 5 ijerph-18-11441-t005:** Summary of findings for the comparison: LI-BFR versus RT alone on muscle mass (cm^2^).

Resistance Training with Blood Blow Restriction Versus Resistance Training					
**Population:** Non-active older adults **Intervention:** resistance training with blood flow restriction **Comparison:** resistance training **Setting:** laboratory					
**Outcomes**	**Relative Effect (95% CI)**	**Anticipated Absolute Effect *** **(95% CI)**		**N° of** **Participants** **(Studies)**	**Certainty** **of the Evidence** **(Grade)**
		**Assumed Risk with Control**	**Assumed Risk with Intervention**		
**Muscle mass (cm^2^)** Up to 12 weeks	SMD 0.62 * (−0.09 to 1.34)	10.7 to 61.7	Mean strength in intervention was 0.62 higher (0.09 lower to 1.34 higher)	86 (3 RCTs)	⨁◯◯◯ VERY LOW ^1,2,3^
**Muscle mass knee extensors (cm^2^)** Up to 12 weeks	SMD 0.26 * (−0.39 to 0.91)	47.7 to 61.7	Mean strength in intervention was 0.26 higher (0.39 lower to 0.91 higher)	37 (2 RCTs)	⨁◯◯◯ VERY LOW ^1,3^
**Muscle mass knee flexors (cm^2^)** Up to 12 weeks	SMD −0.20 * (−1.06 to 0.66)	23.5	Mean strength in intervention was −0.20 higher (−1.06 lower to 0.66 higher)	21 (1 RCTs)	⨁◯◯◯ VERY LOW ^1,3^
**Muscle mass elbow flexors and extensors (cm^2^)** Up to 12 weeks	SMD 1.65 * (0.75 to 2.54)	10.7 to 12	Mean strength in intervention was 1.65 higher (0.75 lower to 2.54 higher)	28 (1 RCTs)	⨁◯◯◯ VERY LOW ^1,3^

The risk in the intervention group (and its 95% confidence interval) is based on the assumed risk in the comparison group and the relative effect of the intervention (and its 95% CI). cm^2^: Square centimeters; CI: Confidence interval; SMD: Standard mean difference. * Effects size: 0.2 represents a small effect, 0.5 a moderate effect, and 0.8 a large effect [29]. ^1^ Downgraded by one level due to no randomization process; ^2^ Downgraded by one level due to inconsistency; ^3^ Downgraded by one level due to small sample size and wide confidence intervals (imprecision).

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
