# Peer review of "Resistance Training with Blood Flow Restriction Compared to Traditional Resistance Training on Strength and Muscle Mass in Non-Active Older Adults: A Systematic Review and Meta-Analysis"

_ijerph, 2021, doi:10.3390/ijerph182111441_

Round 1
Reviewer 1 Report
Excellent review
I could use a discussion concerning the training protocols in the study and the degree og blood flow restriction
Author Response
Dear Reviewer
The authors would like to thank you very much for your kindness with the manuscript. We had worked hard on it, so we are really appreciated of you favorable opinion.
Your sincerely
Reviewer 2 Report
The present systematic review and meta-analysis aims to determine the effectiveness of LI-BFR versus dynamic high-intensity resistance training on strength and muscle mass in non-active older adults. First of all, the effort made by the authors to enhance the current knowledge of this promising training method in older adults should be acknowledged. Furthermore, the article is well written, logical, and easy to comprehend.
However, I find some points that may be addressed before being considered for publication in Int. J. Environ. Res. Public Health journal:
1) The abstract should be rewritten in order to be more accurate and consistent with the text of the manuscript. Your conclusion needs to be shorter. Do not ask for further research.
2) The objective included in the abstract is not the same than the one stated in the introduction and discussion. This fact detracts from the consistency of the manuscript and should be adjusted.3) The risk of bias across the studies should be presented as Supplemental Material.
4) The discussion reports a prioritization process for outcomes that has not been described and does not appear previously. Please, correct this issue.
Author Response
REVIEWER 2 RESPONSE
First of all, we highly appreciate the technical comments provided by reviewer. These kinds of comments are really constructive and definitely, with the main aim of help us to improve the quality of the present manuscript.
Please note that the changes made are marked in red for easy detection and review.
REVIEWER 2: The reviewer said: “1) The abstract should be rewritten in order to be more accurate and consistent with the text of the manuscript. Your conclusion needs to be shorter. Do not ask for further research.”
- AUTHORS: Following your suggestion regarding the structure of the abstract, and the authors have changed the abstract (deleting futures research and reducing the extension of our conclusion).
REVIEWER 2: The reviewer said: “2) The objective included in the abstract is not the same than the one stated in the introduction and discussion. This fact detracts from the consistency of the manuscript and should be adjusted”.
- AUTHORS: We have readjusted the wording of the objectives both in the abstract and in the discussion so that they follow a common line throughout the document.
REVIEWER 2: The reviewer said: “3) The risk of bias across the studies should be presented as Supplemental Material”.
- AUTHORS: Thanks for the suggestion, however, recently the Cochrane collaboration suggested using the risk of bias tool version 2 and the review team decided to leave the figures to show that we adhere to the most current and robust methods for the development of systematic reviews.
REVIEWER 2: The reviewer said: “4) The discussion reports a prioritization process for outcomes that has not been described and does not appear previously. Please, correct this issue”.
- AUTHORS: We have proceeded to remove the section of the discussion you mention in your review.